# Offline Imitation Learning
# with a Misspecified Simulator*

**Shengyi Jiang, Jing-Cheng Pang, Yang Yu**
State Key Laboratory of Novel Software Technology,
Nanjing University, Nanjing, China
{jiangsy, pangjc, yuy}@lamda.nju.edu.cn
.

## Abstract

In real-world decision-making tasks, learning an optimal policy without a trial-and-error process is an appealing challenge. When expert demonstrations are available, imitation learning that mimics expert actions can learn a good policy efficiently. Learning in simulators is another commonly adopted approach to avoid real-world trials-and-errors. However, neither sufficient expert demonstrations nor high-fidelity simulators are easy to obtain. In this work, we investigate policy learning in the condition of a few expert demonstrations and a simulator with misspecified dynamics. Under a mild assumption that local states shall still be partially aligned under a dynamics mismatch, we propose imitation learning with horizon-adaptive inverse dynamics (HIDIL) that matches the simulator states with expert states in a $H$-step horizon and accurately recovers actions based on inverse dynamics policies. In the real environment, HIDIL can effectively derive adapted actions from the matched states. Experiments are conducted in four MuJoCo locomotion environments with modified friction, gravity, and density configurations. Experiment results show that HIDIL achieves significant improvement in terms of performance and stability in all of the real environments, compared with imitation learning methods and transferring methods in reinforcement learning.

## 1 Introduction

Reinforcement Learning (RL) [1] has achieved remarkable success in virtual environments like Atari games [2], StarCraft II [3], and Go [4]. The principal commonality shared by these virtual environments (predefined games or man-made simulators) is the access to unlimited training data. Such a virtual trial-and-error makes the learning process not only faster but also safer, since execution failures lead to zero physical damage.

Current approaches to apply RL in real-world decision-making tasks without a costly trial-and-error process can be divided into two major categories. One is Imitation Learning (IL) that obtains a policy by mimicking the behavior of human experts from demonstrations. The other is to train a policy in a simulator and then adapt it to the real world. Both approaches have their own limitations — collecting sufficient expert demonstrations or building a high-fidelity simulator that perfectly recovers real environments are both laborious and expensive. A problem naturally arises: How can we relax the requirements of these two methods yet yield a ready-to-deploy policy. Imagine a feasible scenario that we have a few expert demonstrations and a simulator with misspecified dynamics.

In this paper, we propose a method that learns to imitate experts with a few demonstrations and a simulator with misspecified dynamics in a completely offline manner (i.e. no sampling process

*This work is supported by National Key Research and Development Program of China (2017YFB1001903), NSFC(61876077), and Collaborative Innovation Center of Novel Software Technology and Industrialization. Yang Yu is the corresponding author.

in the real world). This setting has been rarely studied before, except for in [5]. None of the two common paradigms of IL, Behavioral Cloning (BC) [6] or Generative Adversarial Imitation Learning (GAIL) [7] can be directly applied in this task. BC treats policy learning as a supervised learning task on expert data. Although BC works in certain environments and is completely offline, it violates a key assumption of statistical supervised learning by considering past predictions that affect the distribution of future inputs. This intrinsic drawback leads to compounding error [8, 9] during policy execution and tends to take the agent to an incorrect state. GAIL, on the other hand, shows better generalization, but requires sampling in the environment. Empirical results in the previous work [10] also prove GAIL does not work if a dynamics mismatch exists.

Our method tackles this hard problem with two key techniques: (1) Horizon-adaptive Inverse Dynamics (HID) and (2) Policy Optimization under Distribution Constraint (PODC). In the training phase, HID learns from expert data in a supervised manner. K ensemble HID models are also trained to estimate the uncertainty of goal state to guide PODC. PODC learns a simulator policy in the simulator by constraining its state distribution with expert data. The state is re-weighted based on the uncertainty estimated by HID ensemble. Since direct optimization over a state distribution is intractable, we adopt Generative Adversarial Networks (GANs) [11]. The final optimization objective takes a similar form as described in Generative Adversarial Imitation Learning from Observation (GAILfO) [12]. In the deployment phase, the simulator policy runs in the simulator for several steps to generate short-term goals. HID then selects the best goal among them and recovers an action based on the state-goal pair.

We evaluate the efficacy of our algorithm with four continuous-control locomotion tasks from MuJoCo [13]. In our experiments, an expert policy is trained under a modified dynamics (serving as a "real-world" dynamic), where one of the gravity, friction and density configurations is changed. A few trajectories are then collected by the expert policy as a fixed size expert data. The default dynamics, serving as the simulator, is where the imitator's policy is learned. The range of modification is $\{0.5, 1.5, 2.0\}$, which is sufficient to show that our algorithm is effective and robust to a wide range of dynamics mismatch. We show that our method yields much better policies than baseline IL algorithms in all these tasks, leading to a successful transfer of expert skills to an imitator in an environment different from where the expert acts without any sampling process.

## 2 Background

### 2.1 Preliminaries

**Reinforcement learning (RL)**. RL solves sequential decision-making problems by instructing an agent to interact with the environment and optimize the policy. An RL environment is usually modeled as an MDP which is characterized by $\langle \mathcal{S}, \mathcal{A}, \mathcal{R}, \mathcal{T}, \gamma \rangle$, where $\mathcal{S}$ is the state space, and $\mathcal{A}$ is the action space. Given an action $a_t$ at state $s_t$, the next state is determined by the state transition function: $s_{t+1} \sim \mathcal{T}(\cdot | s_t, a_t)$, and reward is governed by reward function: $r_t \sim R(s_t, a_t)$. The optimization objective of RL is to train a policy $\pi$ to maximize the expected discounted accumulated rewards, $\mathbb{E}_\pi[\sum_{t=0}^\infty \gamma^t r_t]$, where $\gamma \in (0, 1)$. In this paper, the dynamics mismatch refers to $\mathcal{T}$ functions of two MDPs being different from each other and the rest remaining the same.

**Imitation learning (IL)**. IL methods aim to train a policy to mimic the expert's behavior [14, 9, 15] with expert demonstrations $\mathcal{D}_E = \{(s_0, a_0), (s_1, a_1) \cdots\}$ and no extra reward signal. There are many ways to utilize expert demonstrations, such as BC which directly maximizes the likelihood of expert actions under the training policy for each state appears in the expert demonstration, and inverse reinforcement learning (IRL) [16, 17, 18] that finds a cost function under which the expert is uniquely optimal. GAIL frames IL as an occupancy-measure matching or divergence minimization problem. It alternatively trains a policy $\pi_\theta$ and a discriminator $D_\omega : S \times A \rightarrow [0, 1]$ to optimize the min-max objective similar to GANs:

$$\min_{\pi_\theta} \max_{D_\omega} \mathbb{E}_{(s,a) \sim \pi_E} \left[\log D_\omega(s, a)\right] + \mathbb{E}_{(s,a) \sim \pi_\theta} \left[\log\left(1 - D_\omega(s, a)\right)\right] - \lambda \mathcal{H}\left(\pi_\theta\right), \quad (1)$$

where $D : \mathcal{S} \times \mathcal{A} \rightarrow [0, 1]$ is a classifier trained to discriminate between the state-action pairs from the expert and those from the imitator, and $\lambda \mathcal{H}(\pi_\theta)$ is the entropy term. The original GAIL approach can be modified to work in the absence of actions. Specifically, Eq.(1) can be altered to

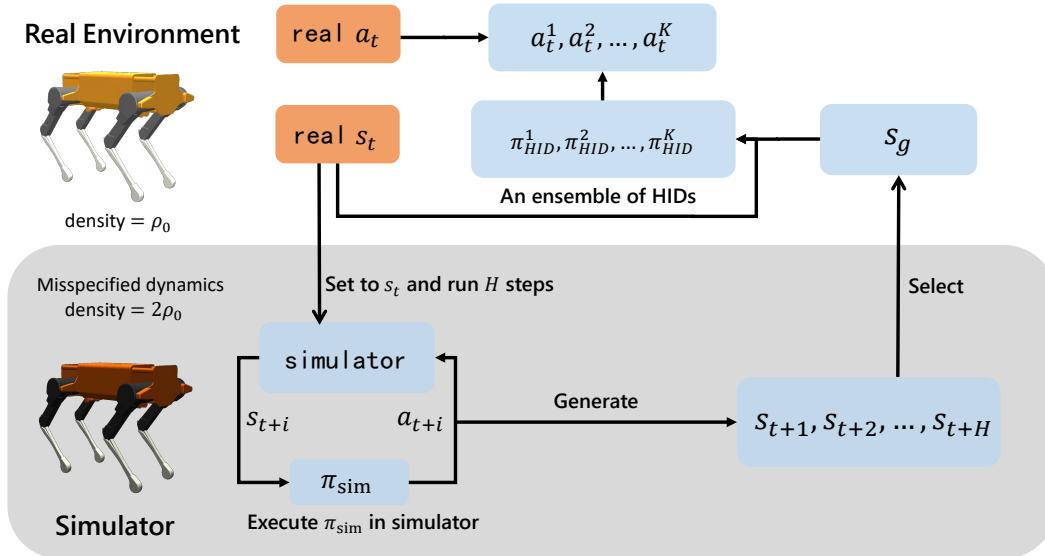

Figure 1: Illustration of the structure of the model and the deployment procedure. The entire model is composed of two parts: one policy is trained and utilized in the simulator, while the other is trained and utilized in the real environment. Algorithm 2 describes the detailed procedure of joint training.

use a state-dependent discriminator $D_\omega(s)$, and state-visitation (instead of state-action visitation) distributions $\rho_E(s)$ and $\rho_{\pi_\theta}(s)$. GAILfO is an example of such method that learns a discriminator $D_\omega : \mathcal{S} \times \mathcal{S} \to [0,1]$ instead. GAIL and GAILfO recover the same policy if converged to the optimum.

Our method contains a variant of GAILfO to learn policy in a simulator. The policy learned with GAIL is not directly deployed, but is used to generate good future states. The fake samples are also re-weighted to perform a partial matching.

**Inverse Dynamics** and **Goal-Conditioned Policy**. Given a state transition tuple $(s_t, a_t, s_{t+1})$, the inverse dynamics takes $(s_t, s_{t+1})$ as input and recovers action $a_t$ to reflect local environment transitions. Most previous works use inverse dynamics to perform curiosity-driven exploration [8, 19, 20]. PCHID [21] learns a k-step policy by directly using Hindsight Inverse Dynamics. Goal-conditioned policy also takes a state pair $(s_t, s_g)$ as input and outputs action $a_t$ to reach the goal state $s_g$ from $s_t$. LfP [22] learns a goal conditioned policy from "playing" with data collected by a human demonstrator, and goalGAIL [23] performs goal-conditioned imitation learning with goal-labeled expert demonstrations. GCSL [24] shows a goal conditioned policy learned in a supervised way can still solve sequential decision tasks. RPL [25] decomposes long-horizon tasks into several sub-tasks that are solvable by k-step local goals.

Our method contains HID which extends one-step future state $s_{t+1}$ in the original inverse dynamics to a set of $H$ future states $\{s_{t+h}|h \in \{1 \ldots H\}\}$. HID regards these near-future states as local goals of the current state. It can be viewed as a hybrid of inverse dynamics and goal-conditioned policy. We name our method "horizon-adaptive inverse dynamics". Experiment results show that our method generalizes better on a fixed expert dataset without a specially designed or relabeled goal state.

## 2.2 Related Work

Two key topics of our work that extend imitation learning are (1) learning with dynamics mismatch; (2) an offline training manner. There are some works tackling dynamics mismatch in IL. A common technique is to constrain the state distribution of agents and experts. I2L [10] extends the GAIL framework to a two-stage optimization problem by first minimizing the distribution shift between the replay buffer and expert demonstrations and then performing conventional GAIL optimization process; SAIL [26] aligns the states visited by agents and experts by minimizing the Wasserstein Distance between them. Such techniques are also used in off-policy reinforcement learning to stabilize the training process. CSDS [27] uses a density estimator to estimate the state distribution and then develops a constrained off-policy gradient objective that minimizes the distribution shift.

---

**Algorithm 1** Deployment Process of HIDIL

---

**Input:** $K$ HID policies $\pi_{\text{HID}}$, simulator policy $\pi_{\text{sim}}$, discriminator $D_\omega$, simulator.
**Output:** Policy $\pi_{\text{real}}$ running in real environment.
   **while** not done **do**
      Observe $s_t$ from real environment.
      Set simulator state to $s_t$.
      Execute $\pi_{\text{sim}}$ in the simulator for $H$ steps and get $\{s_{t+1}, \ldots, s_{t+H}\}$.
      Select $s_g = \text{argmax}_{s_g \in \{s_{t+1}, \ldots, s_{t+H}\}} D_\omega(s_t, s_g)$.
      Output $a_t = \pi_{\text{HID}}(s_t, s_g)$, where $a_t$ is the output closest to the mean of the ensemble models'
      output.
   **end while**

---

Regarding the second topic, BC and its variations [28, 29] are the only methods capable of training policies in the offline setting simply from expert demonstrations. All these methods train policies in a supervised manner. Our work makes the most of expert demonstrations and a simulator with imperfect dynamics and provides a feasible way to imitate the expert's policy in an offline manner. Christiano et al. [5] study offline imitation learning with a simulator similar to our setting. They propose a method that combines an inverse dynamics policy learned with expert demonstrations and another policy generating $s_{t+1}$ by sampling in the simulator. The similarities and differences with their work are elaborated in Sec. 3.

## 3   Imitation Learning with Horizon-adaptive Inverse Dynamics

We study offline imitation learning with expert demonstrations and a misspecified simulator. To validate our method, we configure simulators in both "simulation" and "real-world" environments as the amplitude of variation between "simulation" and "real-world" environments can be easily adjusted. Let $\mathcal{D}_{\text{E}} = \{\{s_0, a_0, ..., s_T, a_T\}_1, \ldots, \{s_0, a_0, ..., s_T, a_T\}_i\}_{\text{E}}$ denote a fixed set of demonstrations collected by experts in the real world. We start by analyzing why previous methods perform poorly under this setting and propose our improved method.

**Motivation**. Christiano et al. [5] propose a method to combine an inverse dynamics policy learned with expert demonstrations and a simulator policy $\pi_{\text{sim}}$ trained in simulator. While in deployment, $\pi_{\text{sim}}$ runs in the simulator to generate the next state, and then the inverse dynamics policy recovers an action. However, their practical algorithm performs worse than expected when expert data is limited or the dynamics change more than a little. Two reasons, according to our analysis, cause this phenomenon: (1) a policy is trained in the simulator from scratch, leaving the expert data alone. The simulator policy will inevitably visit states that are dissimilar from expert data since this policy is trained independently of the expert data. Thus the inverse dynamics policy outputs an improper action when receiving such states. Fig. 2 shows the reward discrepancy between the policy trained in the simulator and the policy

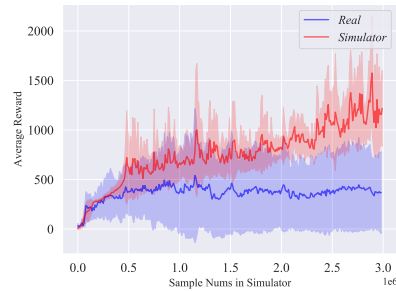

Figure 2: An illustration of reward discrepancy between the policy trained in the simulator and the policy deployed in the real world.

deployed in the real world. Though the policy is trained, it does not provide correct guidance to the inverse dynamics in the real environment, resulting in poor performance; (2) an assumption that 1-step inverse dynamics is similar across simulator and the real world is made. Such assumption usually fails under the dynamics mismatch. We loosen this constraint by introducing the idea of H-step equivalence that two policies in different dynamics can reach the same goal state $s_g$ from any $s_t$ within $H$ steps, as illustrated in Fig. 3. Our method implements these two ideas by designing modules elaborated in the following sections. The key idea of our method is to match states in the simulator to expert states in a $H$ step horizon and accurately recover actions based on inverse dynamics policies.

Fig. 1 shows the overall structure of our method and the workflow in the deployment process. Our method is comprised of two modules, one being a horizon-adaptive inverse dynamics that mimics expert behaviors, the other being a policy $\pi_{\text{sim}}$ trained in the simulator to generate future states. While

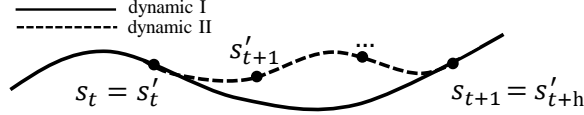

Figure 3: Illustrations of agents in different dynamics takings different steps to reach the same goal.

deploying in the real world, we generate actions to be executed in the following procedure: given state $s_t$ observed in the real environment, we set it as the current state in the simulator and simulate with policy $\pi_{\text{sim}}$ for $H$ steps to get a set of future states $\{s_{t+h}|h \in \{1, \ldots, H\}\}$. Then the agent chooses the best one from them as the goal state $s_g$. The standard for "best" will be discussed later. Finally, the agent in the real world executes actions recovered from inverse dynamics policy. The training and the deployment procedure are elaborated in Alg. 1 and Alg. 2 respectively.

## 3.1 Horizon-adaptive Inverse Dynamics (HID)

Usually, an inverse dynamics policy is trained on $(s_t, s_{t+1}, a_t) \sim \mathcal{D}_{\text{E}}$. We propose horizon-adaptive inverse dynamics (HID) that is trained on extended expert demonstrations $(s_t, a_t, s_{t+h})$, where $h \in \{1, \ldots, H\}$. All inverse dynamics policies with different horizons share one neural network for two reasons. First, it introduces an inductive bias that local actions are similar given the current state and a goal within a horizon of $H$. It helps the goal-conditioned policy to generalize better when $s_t$ or $s_{t+h}$ is somewhat inaccurate. Second, since expert data are very limited in the offline setting, training different networks for different horizons or using a more sophisticated network structure may incur severe overfitting. The optimization objective is formalized as:

$$\mathcal{L}(\theta, \mathcal{D}) = \mathbb{E}_{(s_t, s_{t+h}, a_t) \sim \mathcal{D}_{\text{E}}} \|\pi_\theta(\cdot|s_t, s_{t+h}) - a_t\|_2^2. \tag{2}$$

## 3.2 Uncertainty Estimation from HID Ensemble

Expert data can be further exploited to decide if a given state pair is located inside the distribution of expert data by estimating its uncertainty. Consider that expert demonstrations in our case are fixed and limited, a simple but practical approach to estimate the uncertainty is to use an ensemble of several HIDs with different initialization. The variance over the HIDs should be high outside the experts' distribution, since the data is sparse, but low inside the experts' distribution since the data is dense [30]. This can help $\pi_{\text{sim}}$ generate future goal states with low uncertainty. The concrete usage is described in the next subsection. The variance of the output of the ensembled policies' is written as:

$$\text{Var}(\pi_{\text{HID}}(a_t|s_t, s_{t+h})) = \frac{1}{K} \sum_{k=1}^{K} \left( \pi_{\text{HID}}^k(a_t|s_t, s_{t+h}) - \frac{1}{K} \sum_{K=1}^{K} \pi_{\text{HID}}^k(a_t|s_t, s_{t+h}) \right)^2. \tag{3}$$

Since we have $H$ horizon-adaptive inverse dynamics policies, we select the action closest to the mean value as the action to be executed during the deployment phase.

## 3.3 Policy Optimization under Distribution Constraint

Following the aforementioned motivation, the target for $\pi_{\text{sim}}$ is to generate state-goal pairs that resemble expert data. Directly minimizing the distance between two complex distributions is intractable. However, $f$-GAN [31] proves that generative-adversarial training is equivalent to minimizing certain kinds of distances between two distributions. A training objective that resembles the form of GAILfO can be written as:

$$\min_{\pi_\theta} \max_{D_\omega} \mathbb{E}_{(s,s') \sim \pi_\theta} \left[ \log \left( D_\omega \left( s, s' \right) \right) \right] + \mathbb{E}_{(s,s') \sim \pi_{\text{E}}} \left[ \log \left( 1 - D_\omega \left( s, s' \right) \right) \right]. \tag{4}$$

Note the entropy term is omitted here since we do not need the policy itself, but the good goal states. The generator is implicit in the form of a policy interacting with the environment and collecting

---

**Algorithm 2** Training Procedure of HIDIL

---

**Input:** Expert demonstrations $\mathcal{D}_{\mathrm{E}}$, number of ensemble policies $K$, horizon $H$, simulator.
**Output:** $K$ HID policies $\pi_{\mathrm{HID}}$, discriminator $D_\omega$, simulator policy $\pi_{\mathrm{sim}}$.
  Initialize $K$ horizon-adaptive inverse dynamics policies $\pi_{\mathrm{HID}} = \{\pi_{\mathrm{HID}}^1, \ldots, \pi_{\mathrm{HID}}^K\}$.
  // Train $\pi_{\mathrm{HID}}$
  Generate $\hat{\tau}_{\mathrm{exp}}$ by relabelling $(s_t, a_t) \sim \tau_{\mathrm{exp}}$ to $(s_t, a_t, s_g)$ where $s_g \sim \{s_{t+1}, \ldots, s_{t+H}\}$.
  **for** $k = 1 \to K$ **do**
    **while** not converged **do**
      Sample $B$ from $\hat{\tau}_{\mathrm{exp}}$.
      Train $\pi_{\mathrm{HID}}^k(a_t|s_t, s_g)$ on $B$.
    **end while**
  **end for**
  // Train $\pi_{\mathrm{sim}}$
  Initialize $\pi_{\mathrm{sim}}$.
  **while** not converged **do**
    Run $\pi_{\mathrm{sim}}$ in simulator and collect trajectories $\tau_{\mathrm{pol}}$.
    Generate $\hat{\tau}_{\mathrm{pol}}$ by relabelling $(s_t, a_t) \sim \tau_{\mathrm{pol}}$ to $(s_t, a_t, s_g)$ where $s_g \sim \{s_{t+1}, \ldots, s_{t+H}\}$.
    **for** each discriminator update iteration **do**
      Sample $B_{\mathrm{pol}}$ from $\hat{\tau}_{\mathrm{pol}}$ and $B_{\mathrm{exp}}$ from $\hat{\tau}_{\mathrm{exp}}$.
      Update $D_\omega$ using Eq.(5).
    **end for**
    Compute trajectory reward $r_t = \max_{s_g \sim \{s_{t+1}, \ldots, s_{t+H}\}} \log D_\omega(s_t, s_g)$.
    Update $\pi_{\mathrm{sim}}$ using PPO.
  **end while**

---

samples. Recalling our claims on the $H$-step equivalence of policies, we are extending the input to $H$ state-goal pairs for every $s$.

$$\min_{\pi_\theta} \max_{D_\omega} \mathbb{E}_{(s_t, s_{t+h}) \sim \pi_\theta} \left[ \sum_{h=1}^{H} w_h \log\left(D_\omega\left(s_t, s_{t+h}\right)\right) \right] + \mathbb{E}_{(s, s_h) \sim \pi_{\mathrm{E}}} \left[ \sum_{h=1}^{H} \log\left(1 - D_\omega\left(s, s_h\right)\right) \right].$$

(5)

We want $\pi_{\mathrm{sim}}$ to generate goal states that are less uncertain. Unlike previous work [32] that simply adds the variance of ensemble models' output to reward, we utilize the variance to re-weight the data generated by $\pi_{\mathrm{sim}}$. It can be interpreted in the way that the discriminator mainly focuses on the pairs that are within the expert distribution (i.e., with low variance (uncertainty)), rather than the full set $\{(s_t, s_{t+h}), h \in \{1, \ldots, H\}\}$. Such relaxation allows the policy to reach a good goal state via some bad states in different dynamics. The weight $w_h$ is computed via a relaxed Softmax function on the negation of the variance of $(s_t, s_{t+h})$ pair as equation 6 shows:

$$w_h = \sigma\left(z/\mathrm{T}\right) = \frac{\exp\left(-\operatorname{Var}(\pi_{\mathrm{HID}}(a_t|s_t, s_{t+h}))/\mathrm{T}\right)}{\sum_{h=1}^{H} \exp\left(-\operatorname{Var}(\pi_{\mathrm{HID}}(a_t|s_t, s_{t+h})/\mathrm{T}\right)},$$

(6)

where $\mathrm{T}$ is a hyper-parameter on the amplitude of softening the weight on the maximum goal state $s_g$. This operator falls back to $\mathrm{mean}$ when $T_i \to \infty$. For $T_i = 1$, the original Softmax operator is obtained. We use a relaxed Softmax to focus on the most relevant (or most correct) state-goal pairs.

Once the discriminator is fitted, the policy can be trained with any model-free RL algorithm on the reward $\max_{h \in \{1, \ldots, H\}} \log D_\omega(s_t, s_{t+h})$. In our case, we choose Proximal Policy Optimization (PPO) [33] as the policy optimization algorithm.

## 4 Experiments

In this section, we evaluate our method in the MuJoCo physics simulator from OpenAI Gym. Dynamics mismatch is constructed by modifying the configuration files of MuJoCo. For reasons elaborated in Sec. 3, we choose MuJoCo as both simulation and real-world environments. Specifically, the default configuration is regarded as the simulation environment ($\mathbf{E}_{\mathrm{sim}}$) and the modified environment is regarded as the real-world environment ($\mathbf{E}_{\mathrm{real}}$). We have ensured that none of the modifications

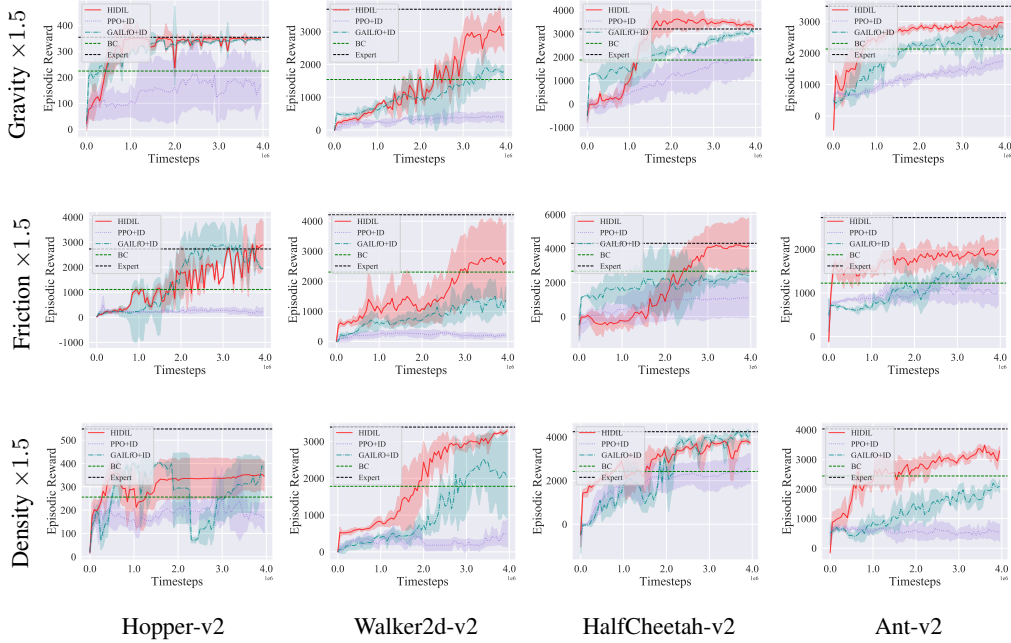

Figure 4: Training curve of different methods when dynamics variety amplitude is 1.5.

changes the observation space or action space. Experiment results will be shown in the three following themes:

1) training process in different MuJoCo environments with various dynamics;
2) state distributions of our simulator policy in $\mathbf{E}_{\text{sim}}$ and expert policy in $\mathbf{E}_{\text{real}}$;
3) connection between the variance of ensemble's outputs and the performance of HIDIL in $\mathbf{E}_{\text{real}}$.

### 4.1 Experiment Setting

In MuJoCo environments, gravity, friction, and density are the three configurations that influence environment dynamics without changing the observation space. In our experiments, we modify one of them each time with three levels of amplitude: $\{0.5, 1.5, 2.0\}$. That is, the $\{$gravity, friction, density$\}$ of $\mathbf{E}_{\text{real}}$ is the coefficient of $\mathbf{E}_{\text{sim}}$ respectively. In all of our experiments, 10 expert demonstration trajectories collected by a stochastic expert policy are given to all algorithms as expert demonstrations. Short trajectories are omitted to ensure the performance of expert demonstrations is above a certain threshold. Every method trained in the simulator samples 4M timesteps in $\mathbf{E}_{\text{sim}}$. The horizon $H$ in HIDIL is set to 5 and $T$ is set to 1 by default across all tasks and dynamics configurations. HIDIL is not sensitive to the setting of horizon and results are shown in Appendix A.3. The methods that require supervised training, i.e. HID, are all trained until convergence. All results reported are averaged with 3 random trials.

To the best of our knowledge, [5] is still the state-of-the-art method under this setting. Since there is not an open-source code of their method, we do our best to reproduce their results and compare HIDIL with it, along with a simpler version of HIDIL and a BC baseline. These methods are named and concisely described as follows. 1) **PPO+ID**: [5] trains a policy in simulator and recovers an action from an inverse dynamics policy; 2) **GAILfO+ID**: a simpler version of HIDIL taking one-step $(s_t, s_{t+1})$ only, which trains a policy in simulator using original GAILfO and recovers an action from an inverse dynamics policy; 3) **BC**: baseline method which trains a policy in a supervised manner based on expert $(s_t, a_t)$ pair. For brevity, we use $\pi_{\text{sim}}$, $\pi_{\text{PPO}}$ and $\pi_{\text{GAILfO}}$ to represent the simulator policy in HIDIL, PPO+ID, and GAILfO+ID respectively.

### 4.2 Experiment Results

**Episodic reward of the deployed policy** $\pi_{\text{real}}$. Fig. 4 shows the curves of the episodic reward of the deployed policy $\pi_{\text{real}}$ as the training of $\pi_{\text{sim}}$ progresses. We present results under the amplitude

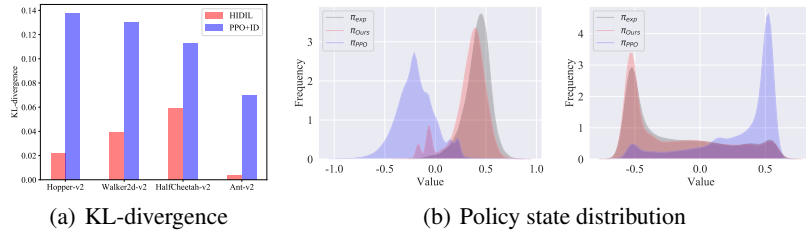

| (a) KL-divergence | (b) Policy state distribution |

Figure 5: Analysis of policy distribution similarity over $\pi_{\text{Ours}}$ and $\pi_{\text{PPO}}$. (a). Estimated KL-divergence when friction $\times$ 0.5. (b). Visualization of one dimension of state distributions over policies following different training manners.

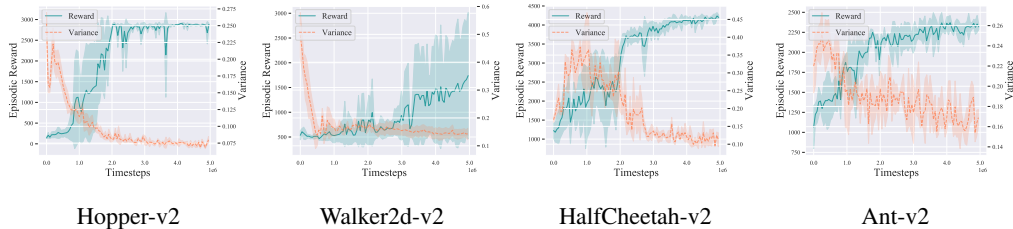

Hopper-v2          Walker2d-v2          HalfCheetah-v2          Ant-v2

Figure 6: Correlation between variance of ensemble models' outputs and $\pi_{\text{real}}$'s performance.

of 1.5 for each type of dynamics mismatch. HIDIL behaves significantly better than the baseline methods. GAILfO+ID also shows considerable results in some environments but suffers from high variance and instability. It helps prove that even the simplest way to constrain the state distribution in the training phase of a simulator could yield a passable policy. PPO+ID fails to imitate expert's policy in most environments except for HalfCheetah, probably because agents in HalfCheetah can always run to the maximum timestep no matter how poorly they behave, thus bad states incur smaller damage to the overall performance. HIDIL degrades into GAILfO+ID in some experiments in which the goal state can be reached within 1-step, which reflects the "adaptivity" of HIDIL. Table 1 shows the normalized scores of 4 methods in MuJoCo environments based on the final score when training stops. HIDIL reaches a average of $78.1\%$ of expert performance over all kinds of modifications and environments while other methods are all below $65\%$. Since the expert fails to perform well under some dynamics modifications in $\mathbf{E}_{\text{real}}$ (e.g. Gravity $\times 2$ in Hopper-v2), the experiment results in those environments are not taken into account. Experiment results of other levels of modification amplitude can be found in Appendix A.1.

**Analysis of state distribution similarity**. We estimate state distribution similarity by calculating KL divergence between state distribution of our policy and the expert. Since the original KL-divergence between policy state distributions is intractable, we simplify the calculation by estimating KL-divergence independently in every dimension. Fig. 5 shows the numerical results of policy state distribution similarity. Both numerical results on average KL-divergence and visualization on two selected dimensions show states generated by our method align with expert data better.

**Analysis of the correlation between variance and reward**. Fig. 6 corroborates such a correlation between the variance of the output of ensemble policies and the episodic reward of the deployed policy during the training process. In all cases, the reward improves while the uncertainty decreases. The results show that the deployed policy can reach a higher reward and be more stable if the final variance is smaller. In some cases, we also find that a sudden rise in the variance indicates a drop in the final performance, which proves to be an effective criterion to early-stop the training process in the simulator.

Ablation studies on sample re-weighting are shown in Appendix B.

Table 1: Normalized scores over all experiments.

| Method | HIDIL | GAILfO+ID | PPO+ID | BC | Expert |
|---|---|---|---|---|---|
| Normalized Score | **78.1%** | 64.8% | 32.8% | 55.0% | 100.0% |

# 5 Conclusions

In this paper, we propose HIDIL to solve an offline imitation-learning problem, in which a few expert demonstrations and simulator with misspecified dynamics are given. We propose an idea of imitation learning with horizon-adaptive inverse dynamics that matches state distributions for up to $H$ steps. To fully utilize expert demonstrations in the offline setting, we introduce an ensemble model to help estimate the uncertainty and stabilize the training process. Experiment results in four locomotion environments with modified friction, gravity, and density configurations show that HIDIL achieves significant improvement in the real-world environment, compared with current imitation learning and transfer RL methods. Based on the results of our work, we would like to point out that combining expert data and an imperfect simulator in the offline setting is a promising idea to deploy RL in real-world decision-making tasks and deserves more attention.

## Broader Impact

In this paper, we present an offline imitation-learning approach that utilizes a few expert demonstrations and a simulator with misspecified dynamics. Potentially, it can be used to improve the performance of sim-to-real transfer without dangerous exploration in the early training stage of conventional algorithms. Since RL has not been applied in complex real-world tasks, the ethical concerns of this work are very limited. The major drawback is that the final policy $\pi_{real}$ may still produce undesirable actions or reach bad states. In the deployment phase, the agent should be monitored carefully. Other researches on safe reinforcement learning may also help alleviate this problem.

## Acknoledgement and Funding Disclosure

We thank Feng Xu and Fan-Ming Luo for their helpful feedback on the manuscript. This work is supported by National Key Research and Development Program of China (2017YFB1001903), NSFC(61876077), and Collaborative Innovation Center of Novel Software Technology and Industrialization.

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
