[Supplementary Material]

# A  Full Experiment Results

## A.1  Training Process on All Amplitudes of Variety

Figure 7: Training process in four MuJoCo environments with various types of dynamics mismatch.

## A.2 Empirical Correlation Between Variance and Reward

Figure 8: Full results on the correlation between variance and episodic reward.

## A.3 Comparison between different horizons

Fig. 9 shows full experiment results when $H = 3$ and $H = 5$. In most experiments, the performance under different horizons are about the same, while a larger horizon leads to better performance in some other experiments.

Hopper-v2          Walker2d-v2          HalfCheetah-v2          Ant-v2

Figure 9: Comparison between ($H = 3$) and ($H = 5$).

## A.4 Ablation study on re-weighted GAILfO

We re-weight the discriminator loss of GAILfO and do ablation study on it. Fig. 10 shows the ablation study on the re-weighted discriminator loss of fake data. Re-weighting the discriminator loss leads to better performance.

Hopper-v2          Walker2d-v2          HalfCheetah-v2          Ant-v2

Figure 10: Ablation study on re-weighting.

## B Convergence of GAILfO with/without re-weighting

Figure 11: Discriminator's accuracy on real and fake samples.

It is worth noting that, although the generative training empirically derives a good policy $\pi_{\text{sim}}$, the GAN does not converge in most cases. In MuJoCo environments, the accuracy of the discriminator on real and fake samples converge to around 0.8, which is distant from the desired 0.5, as Fig. 11 shows. Previous results on GAIL also confirm such an observation. It is partially due to the generator being implicitly induced by a policy and can not be optimized directly. Despite HIDIL has reached a good empirical performance, such results suggest that the distance between policy distribution and expert distribution can be further optimized and the general framework proposed in this paper could benefit from more recent advancements in constraining state distributions in Reinforcement Learning.

## C   Implementation Details

| Hyper-parameter | Value |
|---|---|
| Horizon | 5 |
| GAE $\lambda$, $\gamma$ | 0.95, 0.99 |
| Tempreture | 1 |
| Policy network | 4 layers, [128,256,128,64], relu |
| BC, ID network | 2 layers, [100,100], relu |
| BC, ID optimizer, lr | Adam, 1e-4 |
| Ensemble model Nums | 5 |
| Discriminator epoch of GAILfO | 5 |
| Iterations Num. for training BC, ID | 20000 |
| Trajectories Num. of expert demonstrations | 10 |
| PPO optimizer, lr, batchsize, epoch, clip ratios | Adam, 3e-4, 512, 10, 0.2 |