[Reviews · NeurIPS 2020]

Review 1

Summary and Contributions: The paper proposes an algorithm to make learning possible in real world. The algorithm has two main modules, 1. a horizon adaptive inverse dynamics that is trained using expert data in real world, 2. a policy in the simulator that is being trained to imitate the real world expert. The policy in the simulator is trained using imitation from observation to generate states similar to the real world expert and then inverse dynamics is used to generate the actions to be taken in real world.

Strengths: Results show improvements compared to baselines. Changes made to GAILfO show improvement in the IfO results which could be interesting if investigated by itself. The overall method and the claims seem to be sound and the problem that is being addressed is related to imitation learning and sim-to-real which is relevant to NeurIPS community.

Weaknesses: The writing can be improved significantly. It's hard to follow the paper. Some examples: Line 45: s_g is used without mentioning what that is (the whole paragraph is hard to follow with the amount of information that is given till that point). Line 93: The complete name of the algorithm should be mentioned when an abbreviations is being used for the first time Algorithm 1 is vague. Line 161: It is the other way around. Some notations are confusing. For instance different notations are used for the goal state, s_g, s_{t+k}, s_{t+h}. Line 167 says horizon of K but everywhere else says horizon is shown with H. etc.

Correctness: The claims and the method seem to be correct and the experiments and the baselines make sense.

Clarity: The writing can be improved significantly. Mentioned in the weaknesses of the paper.

Relation to Prior Work: The authors have mentioned some of the previous works and how their work is related to those. There are more papers in sim to real and imitation learning that could be discussed.

Reproducibility: No

Additional Feedback:


Review 2

Summary and Contributions: The authors are proposing an improvement on existing approaches for imitation learning of policies for embodied agents. The approach is a hybrid between sim-to-real RL approaches (which require a simulator closely matching the real world) and real world imitation learning approaches such as GAIL. The general idea of the paper is that there is a simulator, which, however is allowed to have a different dynamics than the "real world". In particular, the assumption is that two policies can reach the same goal state from the same starting point within H steps in the real-world. The algorithm is tested on the OpenAI Gym environment, where both the real world and the simulator environment are simulations (with different parametrization).

Strengths: The paper is theoretically well grounded and represents an advance over the state of the art. The empirical evaluation is standard for the type of work (OpenAI Gym / MuJoCo) The proposed algorithm is particularly significant and novel because it tackles a setup which is very important for real world learning: the existence of some demonstrations for a task and an imperfect model of the environment.

Weaknesses: -The proposed deployment algorithm requires running the simulator at every step of the policy running - this is much more complex then what typical policy learning algorithms do, and could be a serious limitation in real world deployments. -While we understand that the adaptation of the different dynamics in two simulators is convenient, the paper would be much stronger if the authors would break out from the world of MuJoCo and try out their ideas in the real world.

Correctness: As far as I can tell, the claims, method and evaluation approach are correct.

Clarity: Overall, the paper is very well written. Some comments are below: -In the introduction s_g and s_t are used without definition. -Algorithm 1: does not specify what to do with the different actions returned by the K pi_HID policies. One needs look into section 3.2 to see that what the authors do is to "select the action that is closest to the mean value". This is not at all obvious.

Relation to Prior Work: The paper makes it clear how it relates to previous algorithms (eg GRAILfO), and there is significant novelty.

Reproducibility: Yes

Additional Feedback: I read the feedback which does not change my review.


Review 3

Summary and Contributions: The paper presents an approach to leverage a miss-specified simulator and few expert demonstrations to accelerate learning on a system with a limited interaction budget. The key idea is to extend one step state matching to an H step state matching between the two systems, with the assumption that the state distribution overlaps. Convincing experiments are presented on a simulated system to support the idea

Strengths: Various components of the paper have been explored and studied before. The key contribution of the paper is the idea of horizon adaptive inverse dynamics (HID) which tries to align states that are H step apart and use inverse dynamics to recover actions. Authors show that the multistep equivalence of HID introduces the right inductive biases to constraints dynamics mismatch between the two systems and uses the expert data more efficiently. This overcomes some of the shortcomings of a close previous method [Christiano et al. [5]] they compare against. To validate this claim, the authors show in Figure 5 that the distribution induced from HIDIL is very close to the expert distribution.

Weaknesses: - The cornerstone of the paper is H step state equivalence (HID). This ideas isn't novel and has been successfully explored in multiple works --https://arxiv.org/abs/1903.01973, https://arxiv.org/pdf/1910.11956.pdf, etc

Correctness: The presented results are in simulations and the chosen parameters to induce dynamics mismatch are less representative of real-world scenarios. Mismatch in the real world arises from (a) incorrect modeling (b) unmodelled phenomenons. Some phenomena are more catastrophic than others -- delay, noise, etc. Experiment results, while correct, can be improved to strengthen the claims. A real-world result will be ideal.

Clarity: - Figure 1 can be made more effective, its a little hard to follow - Last term of eq 1 isn't introduced - It will help to clarify early on in the paper the domain in which expert data is gathered. It becomes clear quite late in the paper. - section 4.2 typo - remove 'of' - remove '.' at the end of equations

Relation to Prior Work: - Differenced to Christiano et al. [5] is clearly outlined - Relation of prior work on HID need more work.

Reproducibility: Yes

Additional Feedback:


Review 4

Summary and Contributions: This paper studies the problem of offline imitation learning where the simulator is misspecified and a small set of demonstrations from real environment is provided. Given an observation from real environment, the simulator is set with the observation and the policy is rolled out for a limited horizon. A goal is picked from the visited states and inverse dynamics problem is solved. Generated action is executed in the real environment. Overall contributions of the paper are: - Using multi-step horizon to solve the ID problem which alleviates simulator misspecification - Using uncertainty of the ensemble of HIDs as weights for selecting the next action

Strengths: - Using multi-step horizon ID gives considerable improvement on top of GAILfO in continuous control settings with smooth transitions - Combining an ensemble model by using uncertainty of the HIDs as weights is interesting.

Weaknesses: - Multi-step horizon is used rather naively. Authors assume that H-step into the future, the action will change mildly which is the basis for HID. It is not clear to me under which conditions "two policies in different dynamics can reach the same state s_g from any s_t within H steps" is satisfied. - It is assumed that the state of the simulator can be set with observations from real environment. Other than assuming simulator can be arbitrarily modified, I think this ignores POMDP assumption.

Correctness: Claims are somewhat correct. Line 8 in the abstract is not discussed in the paper and it is not clear if the states are fully observable. Empirical methodology is correct with well-know benchmark results.

Clarity: The paper is somewhat clearly written. I outlined some of the concerns above.

Relation to Prior Work: It is clearly discussed.

Reproducibility: Yes

Additional Feedback: - Could you clarify if the environment reward in simulator is used in training the agent? - Line 211, double 'of'. - In Line 151, recovers --> recover.

[Author Response · NeurIPS 2020]

We thank all the reviewers for the time and expertise invested in these reviews. We have corrected the typos and
grammar mistakes, and rephrased some points/sentences to improve the overall readability of our paper. In the following
sections, we respond to the major concerns of each reviewer respectively to, hopefully, clarify our methods and visions.

**Responses to Review # 1**

**Q: What is the meaning of every notation?** A: We are sorry that some abuse of notations in the paper hinders the
understanding of our method. We have checked the notations to ensure that they are consistent throughout the whole
paper. We would further explain that, generally, we use $K$ to denote the number of ensemble models and $H$ to denote
the maximum horizon. Their corresponding lowercase letter refer to one instance in the set, e.g. $s_{t+h}$ means a state
in $\{s_{t+1}, \ldots, s_{t+H}\}$. Specially, $s_g$ means a "good" state in $s_{t+1}, \ldots, s_{t+H}$ that can be regarded as a goal state (to be
consistent with notations in other goal-conditioned work).

**Q: What is the relationship to other Transfer Learning/Imitation Learning method?** A: This work aims to tackle a problem
that lies in the intersection of Imitation Learning and Transfer Learning (more specifically Sim2Real).We have included
the most relevant works that tackle a similar problem and demonstrated our difference and novelty. We would like to
conduct a complete literature review later to cover the recent works in Imitation Learning and Transfer Learning and
add them to the related work section.
Since there are no major flaws pointed out in the review, could the reviewer please raise the overall score?

**Responses to Review # 2**

**Q: Can this method work in real-time control requiring a reset-able simulator?** A: Generally, the computation cost of a
reset-able simulator is comparable to a model-based method and is thus acceptable. We will explore how to relax such a
constraint in future work from both methodological and engineering perspectives.

**Q: Can this method work considering the complexity of real dynamics mismatch?** A: Our empirical results on various
modifications to MuJoCo environments (3 types $\times$ 3 magnitude) can prove that our method is robust to different
dynamics mismatch, so we believe it shows the potential to work in sophisticated real-world problems. We have plans
to apply the method proposed in this work on a real quadruped robot in the future.

**Responses to Review # 3**

**Q: What is the difference compared with others using Goal-conditioned Policy(GCP)/Hindsight Inverse Dynamics(HID)?**
A: HID adopted in our work contributes to the overall purpose to alleviate dynamics mismatch problem and augment the
limited expert data. Other works use GCP/HID for different purposes. PCHID (arXiv 1910.14055) solves goal-oriented
tasks in a supervised manner. Play-GCBC (arXiv 1903.01973) trains a goal-conditioned policy to address a multi-modal
problem. Relay Policy Learning (RPL) mentioned in (arXiv 1910.11956) solves long-term robotic tasks in a hierarchical
manner with the help of a goal-conditioned policy. We will discuss these studies in the related work section.

**Q: Can this method work considering the complexity of real dynamics mismatch?** A: Our empirical results on various
modifications to MuJoCo environments (3 types $\times$ 3 magnitude) can prove that our method is robust to different
dynamics mismatch. So we believe it shows the potential to work in sophisticated real-world problems. We have plans
to apply the method proposed in this work on a real quadruped robot in the future.

**Responses to Review # 6**

**Q: What is the meaning of "partial alignment"** A: We realize that the word "partially" in Line 9 is a little bit confusing
and have clarified it. Instead of referring to environment being "partially observable", it means that not every $(s_t, s_{t+h})$
pair needs to be matched, and a subset (which "partial" actually means) matching is enough. Please refer to Figure 3 in
the paper for a visual illustration. All the experiments are done in fully observable MuJoCo environments.

**Q: What is the rationality behind "partial alignment" assumption?** A: Such an assumption comes from an empirical
observation that in robotics control problems, some key poses in different dynamics are still alike. Aligning such
key poses would make the long-horizon learning much easier. Besides, a weighted GAIL(GAILfO) optimization in
our method is a relaxation to the "exact partial alignment". It can be viewed as an occupancy matching problem that
automatically matches the most "similar" state. We would like to explore further in this area to see if this method will
work under some formalization of dynamics differences.

**Q: Do we use the simulator reward in the training phase?** A: No, we do not use simulator reward functions in any training
phases, i.e., both GAILfO and Goal-conditioned BC (which accords with the usual Imitation Learning setting).

[Meta-Review · NeurIPS 2020]

This paper had a wide spread of reviews and generated significant discussion amongst the reviewers. In the end, the majority of the reviewers agreed that while the main contribution was slightly lacking in novelty (in the sense that it was mostly a retargeting of a known technique to a new setting), it was still a valuable contribution. However, there was not a total consensus, due to R1 having significant concerns about how the paper was written. That said, the majority of reviewers think the paper is strong enough to be accepted, so I recommend that it is, with the caveat that the authors pay close attention to the revision suggestion of R1 to improve the communication of ideas.